# Inhibition of TIGAR Increases Exogenous p53 and Cisplatin Combination Sensitivity in Lung Cancer Cells by Regulating Glycolytic Flux

**DOI:** 10.3390/ijms232416034

**Published:** 2022-12-16

**Authors:** Jiaying Fu, Sihang Yu, Xiyao Zhao, Chaoke Zhang, Luyan Shen, Yanan Liu, Huimei Yu

**Affiliations:** Department of Pathophysiology, College of Basic Medical Sciences, Jilin University, Changchun 130021, China

**Keywords:** TIGAR, p53, cisplatin, apoptosis, A549 cells

## Abstract

The metabolism and apoptosis of tumor cells are important factors that increase their sensitivity to chemotherapeutic drugs. p53 and cisplatin not only induce tumor cell apoptosis, but also regulate the tumor cell metabolism. The TP53-induced glycolysis and apoptosis regulator (TIGAR) can inhibit glycolysis and promote more glucose metabolism in the pentose phosphate pathway. We speculate that the regulation of the TIGAR by the combination therapy of p53 and cisplatin plays an important role in increasing the sensitivity of tumor cells to cisplatin. In this study, we found that the combined treatment of p53 and cisplatin was able to inhibit the mitochondrial function, promote mitochondrial pathway-induced apoptosis, and increase the sensitivity. Furthermore, the expression of the TIGAR was inhibited after a combined p53 and cisplatin treatment, the features of the TIGAR that regulate the pentose phosphate pathway were inhibited, the glucose flux shifted towards glycolysis, and the localization of the complex of the TIGAR and Hexokinase 2 (HK2) on the mitochondria was also reduced. Therefore, the combined treatment of p53 and cisplatin may modulate a glycolytic flux through the TIGAR, altering the cellular metabolic patterns while increasing apoptosis. Taken together, our findings reveal that the TIGAR may serve as a potential therapeutic target to increase the sensitivity of lung cancer A549 cells to cisplatin.

## 1. Introduction

Lung cancer is one of the most common malignant tumors in the world, with the second highest incidence (11.4%) and the highest mortality (18%) in the world [1]. The World Health Organization divides lung cancer into two types according to histology: small-cell lung cancer and non-small cell lung cancer, of which non-small-cell lung cancer is the most common type, accounting for approximately 80–85% of all lung cancer cases [2]. Current treatments for non-small-cell lung cancer include surgery, chemotherapy, and targeted therapy. However, radical surgery cannot be performed on advanced patients, and targeted therapy is only suitable for a small number of non-small-cell lung cancer patients with specific genetic patterns. Therefore, chemotherapy is still the main method for the treatment of non-small-cell lung cancer, and cisplatin is the leading medication among platinum-based anticancer drugs for clinical chemotherapy and the comprehensive treatment of lung cancer [3]. In recent years, some progress has also made in gene therapy for tumors [4,5], as one of the most widely studied tumor suppressor genes, p53, has been shown to be closely related to tumor growth and apoptosis. In terms of clinical treatment, Roth et al., in 1996, used a sub-bronchial direct injection of wild-type p53 recombinant retrovirus suspension to treat nine patients with non-small-cell lung cancer who failed the conventional treatment; in three cases, the tumor volume decreased significantly after the treatment, and in another three cases, the tumor stopped growing [6]. In addition, cancer is not only a genetic disease, but also a disease of the energy metabolism. Studies have shown that aerobic glycolysis is related to the occurrence, development, and drug resistance of tumors [7]. Therefore, it may be effective to increase the sensitivity of cancer cells to cisplatin from the perspective of the tumor cell metabolism and apoptosis.

The TIGAR (*c12orf5*), as a TP53-regulated gene, was originally discovered by induction by ionizing radiation [8]; subsequently named the TP53-induced glycolysis and apoptosis regulator (TIGAR), the TIGAR gene contains two p53 binding sites, one upstream of the first exon and the other, more efficiently, inside the first intron [9]. As a target of p53, the TIGAR can be transcriptionally regulated by p53, and the human TIGARs expression can be regulated by p53, transactivator p51A (Tap63), and p73 protein (TAp73) [10,11]. Although the nomenclature of the TIGAR is related to p53, its regulatory mechanism is not yet clear, and the expression of the TIGAR in tumor cells is not completely dependent on the regulation of p53.

The protein structure of the TIGAR is highly similar to the structural sequence of 2,6-bisphosphofructokinase in 6-phosphofructokinase-2/2,6-bisphosphofructokinase (PFK-2/FBPase-2), which can hydrolyze intracellular fructose 2,6-bisphosphate. Fructose 2,6-bisphosphate substrate is a strong activator of phosphofructokinase-1 (PFK-1), since TIGAR is able to hydrolyze intracellular fructose 2,6-bisphosphate to fructose-6-phosphate; the activity of PFK-1 is reduced by this, which in turn inhibits the cellular glycolytic pathway. At the same time, the TIGAR can also regulate the pentose phosphate pathway (PPP), so that fructose 6-phosphate enters the pentose phosphate pathway to promote the synthesis of nicotinamide adenine dinucleotide phosphate (NADPH) and 5-phosphoribonucleic acid, thereby exerting an antioxidant effect and reducing reactive oxygen species (ROS)-induced apoptosis [12,13]. The TIGAR is located in the cytoplasm and enters organelles, including mitochondria, endoplasmic reticulum, and the nucleus, under different stress stimuli, to regulate the cellular functions. The TIGAR, after being transported to mitochondria, can interact with HK2 and ATP51A to promote the mitochondrial production and function [14,15]. 

Hexokinase 2 (HK2), as the first rate-limiting enzyme of glycolysis, has attracted much attention in research due to its dual roles in the regulation of the metabolism and apoptosis. The mitochondrial localization of HK2 can play an important role in the metabolic remodeling of tumor cells, and HK2 in mitochondria is closely related to the production of ATP, which contributes to the combination of glycolysis and oxidative phosphorylation [16]. HK2 can also inhibit tumor cell apoptosis by inhibiting changes in the mitochondrial membrane’s permeability. The TIGAR can form a complex with HK2 in mitochondria to increase the activity of HK2, which is different from the TIGARs role as 2,6-bisphosphofructokinase to promote the PPP [14]. Therefore, the role of the TIGAR may be a cross-action of the enzymatic and non-enzymatic functions. We speculate that the TIGAR is involved in the regulation of the apoptosis of tumor cells by p53 and cisplatin.

In order to explore the mechanism of increasing chemotherapeutic drug cisplatin sensitivity from the perspective of the tumor cell metabolism and mitochondrial pathway apoptosis, this study targeted the TIGAR to explore the potential role of p53 and cisplatin in the treatment of lung cancer. In this study, we found that in A549 cells treated with exogenous p53 and cisplatin, the expression of the TIGAR was inhibited, the complex formed by the TIGAR and HK2 on mitochondria was reduced, the PPP was inhibited, glycolysis was increased, the ATP production was reduced and altered the cellular metabolic patterns, and the impaired mitochondrial function increased the sensitivity of A549 cells to cisplatin. Therefore, we believe that the combined treatment of exogenous p53 and cisplatin changed the metabolic pattern of non-small-cell lung cancer A549 cells by regulating the TIGAR, increased the mitochondrial pathway apoptosis, and thus increased the sensitivity to cisplatin.

## 2. Results

### 2.1. Combination Treatment of p53 and Cisplatin Increases Apoptosis of Lung Cancer Cell A549

To investigate the effects of cisplatin and p53 on lung cancer cells, we first examined the viability of A549 and H1299 cell lines after the cisplatin treatment and/or p53 overexpression transfection for 24 h. The results of the MTT assay showed that the overexpression of p53 increased the sensitivity of cells to cisplatin (Figure 1a). The flow cytometry experiments also yielded the same results; that is, compared with the control group, the apoptosis of the cells was increased after the overexpression of p53 and the cisplatin treatment, and the apoptosis was further increased after the combined treatment of p53 and cisplatin (Figure 1b). We then examined the basal expression levels of p53 in A549 cells and H1299 cells (Figure 1c). In subsequent experiments, the A549 cell line with the p53 background as wild-type p53 was selected. In addition, we performed the same treatment in the 16HBE cell lines of bronchial epithelial cells, and the flow cytometry results showed that compared with the cisplatin treatment, the combination treatment of p53 and cisplatin did not cause a further apoptosis of the 16HBE cells (Figure 1d). To investigate the pathway through which this increased sensitivity to cisplatin is caused, we detected the expression levels of Bcl-2 family proteins, and the results showed that after the overexpression of p53 and cisplatin treatment, the ratio of BAX/Bcl-2 was up-regulated, and the protein expression of cytochrome C was increased (Figure 1e–h). Afterwards, we observed the fragmentation of the nucleus by fluorescence microscopy (Figure 1i). Therefore, we believe that the apoptosis induced by p53 and cisplatin is mainly related to the mitochondrial pathway.

### 2.2. Combined Treatment of Cisplatin and p53 Promotes Apoptosis of A549 Cells by Reducing Mitochondrial Function by Inhibiting Pentose Phosphate Pathway

Based on previous results, we believe that the overexpression of p53 increases the sensitivity of A549 cells to cisplatin via the mitochondrial pathway. As an important organelle, mitochondria are responsible for many essential cellular functions, including the energy metabolism, the generation of free radicals, the maintenance of calcium homeostasis, and cell survival and death. Mitochondrial damage and dysfunction are closely related to the occurrence of many diseases. Therefore, we examined mitochondrial dysfunction. The mitochondrial ROS levels were examined after a treatment with cisplatin and/or a p53 transfection for 24 h in the lung cancer A549 cell line (Figure 2a) and showed an increased ROS production at the mitochondrial level. We then examined the mtDNA damage after the cisplatin treatment and/or p53 transfection for 24 h and found that the mtDNA copy number decreased after the combined treatment (Figure 2b). We also examined the level of ATP produced in A549 cells and found that the level of ATP production was reduced after a co-treatment (Figure 2c). The combined treatment of p53 and cisplatin reduced the mitochondrial function, and the mtDNA was damaged; thus, we believe that the apoptosis of the A549 cells may be caused by the reduction in the mitochondrial function, and the damage of the mitochondrial function may be related to the pentose phosphate pathway.

In tumor cells, the pentose phosphate pathway enables tumor cells to escape apoptosis by promoting DNA damage repair. It is an important branch of glycolysis and can produce NADPH and ribose-5-phosphate, which are important precursors for DNA repair, and particularly important for cancer cell proliferation; moreover, ribose-5-phosphate can effectively improve damaged mitochondrial function. Therefore, we wanted to determine whether the mitochondrial damage induced by the combined treatment of p53 and cisplatin was related to the pentose phosphate pathway. G6PD, the first rate-limiting enzyme of the pentose phosphate pathway, when activated, promotes the PPP, increases nucleotide synthesis, and enables an efficient DNA repair. Thence, we examined G6PD and found that both the mRNA level (Figure 2d) and the protein level (Figure 2e) of G6PD decreased after the combined treatment of p53 and cisplatin compared with the cisplatin-treated group, indicating that the pentose phosphate pathway is inhibited. The TP53-induced glycolysis and apoptosis regulator (TIGAR) can play a role in cell survival by increasing the PPP flux; therefore, we speculate that the TIGAR may be a related factor in p53 and cisplatin inducing a reduction in the mitochondrial function by inhibiting the PPP.

### 2.3. Expression of TIGAR in A549 Cells Is Associated with p53 and Cisplatin

We compared the basal expression levels of the TIGAR in A549 cells and H1299 cells, and the results showed that the TIGAR was expressed at a higher level in the A549 cell line (Figure 3a). To explore the potential mechanism of the TIGAR in the metabolism and apoptosis, we detected the mRNA level (Figure 3b) and protein level (Figure 3c) of the TIGAR by qPCR and Western blotting, respectively, after the cisplatin treatment and/or p53 transfection in a lung cancer A549 cell line for 24 h. The expression of the TIGAR was increased after transfection with p53 and a cisplatin treatment compared with the control group; however, after the cisplatin treatment combined with p53, the expression level of the TIGAR decreased compared with the cisplatin treatment. Therefore, the decrease observed in the pentose phosphate pathway after a combined treatment may be caused by the decrease in the TIGARs expression.

### 2.4. Combined Treatment of Cisplatin and p53 Changes the Metabolic Pattern of A549 Cells

In addition to regulating the pentose phosphate pathway, TIGAR also regulates cellular glycolysis. To further determine whether TIGAR also has a regulatory role in glycolysis in A549 cells, we treated cisplatin and/or p53 transfection for 24 h, compared with cisplatin treatment, the glucose consumption (Figure 4a) and lactate production (Figure 4b) in the cisplatin combined p53 group were both increased, which suggests that the decreased TIGARs expression in A549 cells resulted in increased levels of glycolysis. In addition, we increased TIGARs expression by transfection plasmid (Figure 4c) and treated cisplatin and p53 transfection for 24 h in A549 cells with increased TIGARs expression levels, the glucose consumption (Figure 4d) and lactate production (Figure 4e) in were both reduced after the increase in the TIGARs expression. Moreover, flow cytometry experiments showed that the increased TIGARs expression reversed the level of apoptosis due to the combination therapy in A549 cells treated with combined p53 and cisplatin (Figure 4g,h). We believe that the co-treatment of p53 with cisplatin changes the metabolic pattern of cells by the downregulation of the TIGARs expression and increased the cell apoptosis. 

### 2.5. Co-Treatment with Cisplatin and p53 Reduces the Localization of TIGAR and HK2 Complexes on Mitochondria

According to the above results, it is speculated that p53 combined with cisplatin can affect glycolysis by regulating the TIGAR, HK2 plays a key role as the rate-limiting enzyme of glycolysis, and HK2 binds to the mitochondrial outer membrane through the N-terminal mitochondrial binding domain, which plays a key role in tumorigenesis, progression, and chemosensitivity. Another study reported that the TIGAR can form a complex with HK2, resulting in increased HK2 activity. To further explore the mechanism of the TIGAR and HK2 in glycolysis, we used immunoprecipitation assays to detect the binding of the TIGAR to HK2 in A549 cells after the cisplatin treatment and/or p53 transfection for 24 h (Figure 5), compared with the cisplatin-treated group; the binding decreased after the combined treatment, indicating that the combined treatment of cisplatin and p53 could reduce the interaction between the TIGAR and HK2. In addition, the binding of HK2 to VDAC was also reduced after the combined treatment of p53 and cisplatin, suggesting that the localization of the complex formed by TIGAR and HK2 on the mitochondria was reduced. Therefore, we believe that the combined treatment of p53 and cisplatin reduces the binding of the TIGAR to HK2 and decreases the amount of their complexes formed on the mitochondria, thereby increasing the apoptosis of A549 cells.

## 3. Discussion

The TIGAR, which was discovered as a p53 target gene, has been gradually recognized in recent years with the gradual development of research, and its duality in tumors has also gradually been recognized; that is, the TIGAR can inhibit the survival and proliferation of tumor cells by reducing glycolysis. At the same time, it can promote the proliferation of tumor cells by inhibiting the accumulation of ROS, preventing tumor cells from apoptosis caused by ROS under stress, and increasing the synthesis of nucleosides and NADPH. Previous studies have shown that the regulation of apoptosis by p53 is closely related to the mitochondrial function and metabolism-related enzymes [17]. In addition, a large number of studies have shown that cisplatin, as a common chemotherapeutic drug, induces cancer cell apoptosis through a complex mechanism [18,19]; however, the use of this drug is prone to drug resistance, so finding ways to increase its sensitivity has become a problem that we need to solve. In this study, we found that apoptosis induced by a combined p53 and cisplatin treatment was closely related to the TIGAR. Therefore, targeting the regulation of the TIGAR in the metabolism and its effect on the mitochondrial function may provide new ideas for increasing the sensitivity of cisplatin therapy.

The role of the TIGAR in tumors is believed to be caused by the cross-action of its enzymatic and non-enzymatic functions. The TIGAR inhibits glycolysis by reducing the PFK-1 activity, and the inhibition of glycolysis increases the pentose phosphate pathway flux and NADPH levels. In addition, the TIGAR also has a non-enzymatic function. The TIGAR inhibits the synthesis of cyclin-dependent kinase (CDK) complex members, resulting in the dephosphorylation of RB [20], and interacts with AKT to phosphorylate AKT and activate AKT, which subsequently mediates the PI3K-AKT-mTOR signaling pathway, thereby promoting tumor growth and metastasis, and inhibiting autophagy and apoptosis [21], as well as NF-kB signaling and inflammatory cytokines [22]. In this study, we found that the reduction in the TIGAR after a combined treatment of p53 and cisplatin altered the metabolic pattern of the tumor cells, and the PPP was inhibited, which is critical for cancer therapy, as a high PPP flux may make cancer cells exhibit a stronger phenotype in terms of proliferation, invasion, and drug resistance. Additionally, previous work in our lab has also demonstrated that the modulation of glycolysis can increase the sensitivity to chemotherapeutics; for example, Xu et al. [23] demonstrated that ABT737 reversed the resistance to cisplatin by targeting the glucose metabolism in human ovarian cancer cells, and Xiang et al. [24] demonstrated that S1 exerts anti-cancer effects in SKOV3 cells by interrupting the glucose metabolism and inducing apoptosis involving the activation of SIRT3, while the combination of the glycolysis inhibitor 2-DG and S1 is able to further enhance the cytotoxicity. Therefore, regulating the inhibition of PPP by the TIGAR after a combined treatment of p53 and cisplatin may help to re-sensitize resistant tumors. Huang et al. [25] also reported that although the inhibition of glycolysis was effective in killing tumor cells, more glucose flux diverted to the PPP increased the NADPH and ribose-5-phosphate production, which are important precursors for nucleotide synthesis and DNA repair, and protect tumor cells from apoptosis. At the same time, our study also found that the reduction in the TIGAR increased the glycolytic flux, which seems to be a powerful phenomenon for tumor cells; however, the decrease in the ATP levels led us to believe that this increase in glycolysis did not adequately replenish the energy needed by tumor cells for survival, so apoptosis inevitably occurred. Therefore, we believe that the combined treatment of p53 and cisplatin modulates the cellular metabolism through the TIGAR, thereby causing apoptosis. Our study confirmed that the increased expression of the TIGAR can reverse the apoptosis by the combination of p53 and cisplatin, but this reversal is limited, and there may be other mechanisms that need to be explored for the regulation of the combination therapy on lung cancer cell apoptosis. In addition, an interesting phenomenon deserves our attention that both the overexpression of p53 and a cisplatin treatment can increase the expression of the TIGAR, but the expression of the TIGAR was inhibited after a combined treatment. Additionally, the difference between the protein and mRNA level of the TIGAR may be affected by various regulatory mechanisms, such as transcriptional level regulation, post-transcriptional regulation, and post-translational regulation. The mechanism needs to be further explored in subsequent experiments.

The abnormal mitochondrial function is involved in the regulation of various pathophysiological processes, and its abnormality is closely related to the proliferation, metastasis, and survival of tumor cells. Veronica et al. [26] believed that the mitochondria were involved in a cisplatin resistance and were related to mitochondrial DNA, mitochondrial dynamics, and mitophagy. For example, Wandee et al. [27] showed the chemosensitizing activity of metformin in combination with cisplatin in cholangiocarcinoma, an effect associated with increased oxidative stress associated with mitochondrial dysfunction and the initiation of cell death. In this study, we found that p53 and cisplatin inhibit the mitochondrial function. HK2 acts as the first rate-limiting enzyme of glycolysis and also helps tumor cells escape apoptosis by localizing itself on the outer membrane of the mitochondria. The promoter of HK2 contains the p53 response element [28]. Wang et al. [29] showed that the deletion of p53 enhanced the stability of the HK2 mRNA by inhibiting miRNA-143 biosynthesis. Han et al. [30] found that the p53-PDK1-HK2 axis promotes chemoreactivity in epithelial ovarian cancer. Xue et al. [31] observed that p53 was able to disrupt HK2 binding to VDAC. Furthermore, Wen et al. [32] showed that LncRNA-SARCC could sensitize osteosarcoma to cisplatin by targeting HK2-mediated glycolysis. Mansour et al. [33] showed that arsenic trioxide combined with cisplatin can synergistically inhibit the activity of hexokinase in Ehrlich ascites carcinoma cells. The subcellular localization of HK2 is particularly important for its function. HK2 localized in mitochondria acts as the rate-limiting enzyme of glycolysis, while HK2 localized in cytoplasm promotes the pentose phosphate pathway and glycogen synthesis pathway [34]. In this study, we also found that the combined treatment of p53 and cisplatin was able to reduce the interaction of the TIGAR and HK2, thereby reducing the mitochondrial function and increasing apoptosis in A549 cells.

As a transcription factor, the type of p53 should also be explored. In this study, the p53 background of the A549 cell line was wild-type p53. p53 is frequently mutated in human tumors. In addition to the loss of the original tumor-suppressive function, tumor-associated mutant p53 proteins often acquire new tumorigenic activities. In metabolic regulation, wild-type p53 inhibits glycolysis, whereas mutant p53 promotes glycolysis through different mechanisms [35]. Zhang et al. [36] found that the inhibition of glycolysis in tumor cells greatly impaired the role of mutant p53 in promoting tumorigenesis. Additionally, Jiang et al. [37] reported that tumor-associated p53 mutants lack G6PD inhibitory activity and that the enhanced PPP glucose flux due to the activation of p53 may increase the glucose consumption and direct glucose to biosynthesis in tumor cells. Therefore, whether mutant p53 has the ability to affect the TIGARs expression, thereby promoting tumorigenesis, should be considered and investigated. This contributes to developing our understanding of the mechanism of action of the TIGAR in the tumor metabolism. In addition, studies have shown that the knockdown of the TIGAR increases the sensitivity of various chemotherapy drugs such as cisplatin, 5-FU, and temozolomide [21,25,38]. Increasing evidence shows that the TIGAR is an important regulator of tumor development and sensitivity to chemotherapy, radiotherapy, targeted therapy, and endocrine therapy. Therefore, the TIGAR is expected to become a potential target for anti-tumor therapy. However, there are still unresolved issues in this study. In A549 cells, p53 and cisplatin did not regulate the expression of HK2, but the binding of HK2 and the TIGAR was reduced, the reasons and mechanisms behind which are still unclear, and further research is needed.

In conclusion, we found that p53 combined with a cisplatin treatment reduced the expression of the TIGAR, the pentose phosphate pathway was inhibited, the glucose flux shifted more towards glycolysis, and the metabolic pattern of A549 cells changed. We also found that the combined treatment of exogenous p53 and cisplatin reduced the binding of the TIGAR and HK2, resulting in a decreased mitochondrial function, mtDNA damage, and ultimately apoptosis (Figure 6). Therefore, the TIGAR may be a key molecule in determining cell fate. Targeting the TIGAR may elucidate the mechanism by which the combined treatment of p53 and cisplatin increases the sensitivity of tumor cells from the perspective of the metabolism and apoptosis, and may provide a theoretical basis for the treatment of non-small-cell lung cancer.

## 4. Materials and Methods

### 4.1. Reagents and Antibodies

Cisplatin (MedChemExpress, New Jersey, USA). 3-(4,5-dimetrylthiazol-2-yl)-2,5-diphenyltetrazolium bromide (MTT) (Sigma-Aldrich, St. Louis, MO, USA). 

The following antibodies were used: anti-Bax(50599-2-Ig), anti-Bcl-2 (12789-1-AP), anti-Cytochrome c(10993-1-AP), anti-p53(10442-1-AP), anti-HK2(22029-1-AP), anti-VDAC1(55259-1-AP), anti-β-actin(66009-1-Ig), peroxidase-conjugated AffiniPure goat anti-mouse IgG (H + L) (SA00001-1), peroxidase- conjugated AffiniPure goat anti-rabbit IgG (H + L) (SA00001-2) (Proteintech, Chicago, IL, USA); anti-G6PD (sc-373886) and anti-TIGAR (sc-166290) (Santa Cruz, CA, USA).

### 4.2. Plasmid and Transfection

Plasmid-overexpressing human p53 and TIGAR were constructed by the Public Protein/Plasmid Library company (Beijing, China). The plasmids were transfected using the Lipofectamine 2000 reagent (Thermo Fisher Scientific, Waltham, MA, USA). Take a 6-well plate as an example and add 4 µg plasmids to each well.

### 4.3. Cell Culture Conditions

The A549, H1299, and 16HBE cells were obtained from the Chinese Academy of Medical Sciences (Beijing, China). The cells were grown in a D-MEM/F12 and RPMI-1640 culture medium (Gibco Life Technologies, Carlsbad, CA, USA) supplemented with 10% fetal bovine serum (Invitrogen, Carlsbad, CA, USA) at 37 °C at a 5% CO_2_ concentration in an incubator.

### 4.4. Cellular Viability Assays

The cell viability was detected using the MTT assay. The cells were first seeded in 96-well plates, and after 24 h of treatment, the MTT reagent was added to the culture for 4 h, DMSO was added to dissolve the formazan, and then the absorbance at 570 nm was detected using a microplate reader (Bio-Rad, California, CA, USA).

### 4.5. Flow Cytometry Analysis

The cell death was analyzed using Annexin-V FITC/PI (BD Biosciences, Franklin Lakes, NJ, USA) staining. Exponentially growing A549 and H1299 cells were seeded in 6-well culture plates at a density of 5 × 10^5^ cells/well. After an exposure to different experimental conditions, the cells were harvested and subjected to Annexin-V FITC/PI staining according to the manufacturer’s instructions. Apoptosis was studied by Guava^®^ easy-Cyte™ flow cytometry (Luminex, Austin, TX, USA).

### 4.6. Western Blot Analysis

After the treatment, the cells were lysed using an RIPA lysis buffer and the protein content was determined using Bradford reagent (Beyotime Biotechnology, Shanghai, China). Afterwards, Western blotting was performed. The samples were separated using 10% (*w/v*) sodium dodecyl sulfate (SDS)-polyacrylamide gel electrophoresis to separate the proteins. Electrophoresis was performed at 80 V for 2 h. The proteins were transferred to PVDF membranes at 100 V for 1 h. Afterwards, the membrane was blocked with 5% (*w/v*) non-fat dry milk at room temperature for 2 h, the corresponding antibody was added and incubated overnight at 4◦C, and it was incubated with the secondary antibody for 2 h at room temperature. Finally, the proteins were visualized using ECL reagents (YEASEN, Shanghai, China) and a gel imager (Syntyos, Cambridge, MA, USA).

### 4.7. Immunofluorescence and Microscopy

Place the coverslips in 24-well plates and seed the cells (5 × 10^4^ cells/well) with different experimental conditions. The cells were fixed with 4% (*w/v*) paraformaldehyde for 20 min, then stained with Hoechst 33342 (1 ug/mL) for 5 min and washed 3 times with PBS. The mitochondrial ROS were measured by staining with the mitochondrial superoxide indicator MitoSOX™ Red (Invitrogen™, Carlsbad, CA, USA) for 20 min. The images were acquired by an Olympus FV1000 confocal laser microscope (Olympus, Tokyo, JPN).

### 4.8. Relative Quantitative Real-Time PCR

The total cellular RNA and total mtDNA were extracted according to the manufacturer’s protocol and reverse transcribed to generate cDNA, which was then detected by a quantitative real-time PCR (RT-qPCR). All the primers for the real-time fluorescent quantitative PCR (RT-qPCR) were constructed by Sangon Biotech (Beijing, Shanghai). p53: 5′-TGTATGCCTCTGGTCGTACC-3′(forward) and 5′-CAGGTCCAGACGCAGGATG-3′(reverse). TIGAR: 5′-ATCCTGAAAGAAGCGGATCAAAA-3′(forward) and 5′-AACTAAGACACTGGCTGCTAATC-3′(reverse). G6PD: 5′-CGAGGCCGTCACCAAGAAC-3′(forward) and 5′-GTAGTGGTCGATGCGGTAGA-3′(reverse). ND1: 5′-CACCCAAGAACAGGGTTTGT-3′(forward) and 5′-TGGCCATGGGATTGTTGTTAA-3′(reverse). 18SrRNA: 5′-TAGAGGGACAAGTGGCGTTC-3′(forward) and 5′-CGCTGAGCCAGTCAGTGT-3′(reverse).

### 4.9. Glucose and Lactate Concentration Measurement

The cells were seeded in 6-well plates at 7.5 × 10^5^ cells/well. After a treatment with different experimental conditions, the cell culture medium was collected, and the protein was extracted and quantified at the same time. The concentrations of glucose and lactate in the culture medium were then determined using a glucose assay kit (RsBio, Shanghai, China) and a lactate assay kit (Jiancheng Bio, Nanjing, China) according to the instructions. The calculation formula is as follows: glucose consumption = glucose concentration (fresh complete medium)–glucose concentration (experimental group), normalized by the protein concentration.

### 4.10. ATP Concentration Determination

The cells were seeded in 6-well plates at 7.5 × 10^5^ cells/well. After the treatment with different experimental conditions, the cell culture medium was collected, and the protein was extracted and quantified at the same time. The ATP levels were determined using an ATP Assay Kit (Beyotime Biotechnology, Shanghai, China) according to the instructions.

### 4.11. Immunoprecipitation

The pretreated samples were incubated with an antibody and protein G agarose beads (Beyotime Biotechnology, Shanghai, China) for 24 h. The protein sample was extracted from the supernatant after centrifugation. The protein sample was analyzed by Western blotting.

### 4.12. Statistical Analysis

The Mann–Whitney test was used to compare the data between the two groups. The Kruskal–Wallis test was used for multiple comparisons. *p* < 0.05 was considered to be statistically significant. All the experiments were repeated 3 times. Statistical analysis was performed using GraphPad Prism 8 (La Jolla, CA, USA).

## Figures and Tables

**Figure 1 ijms-23-16034-f001:**
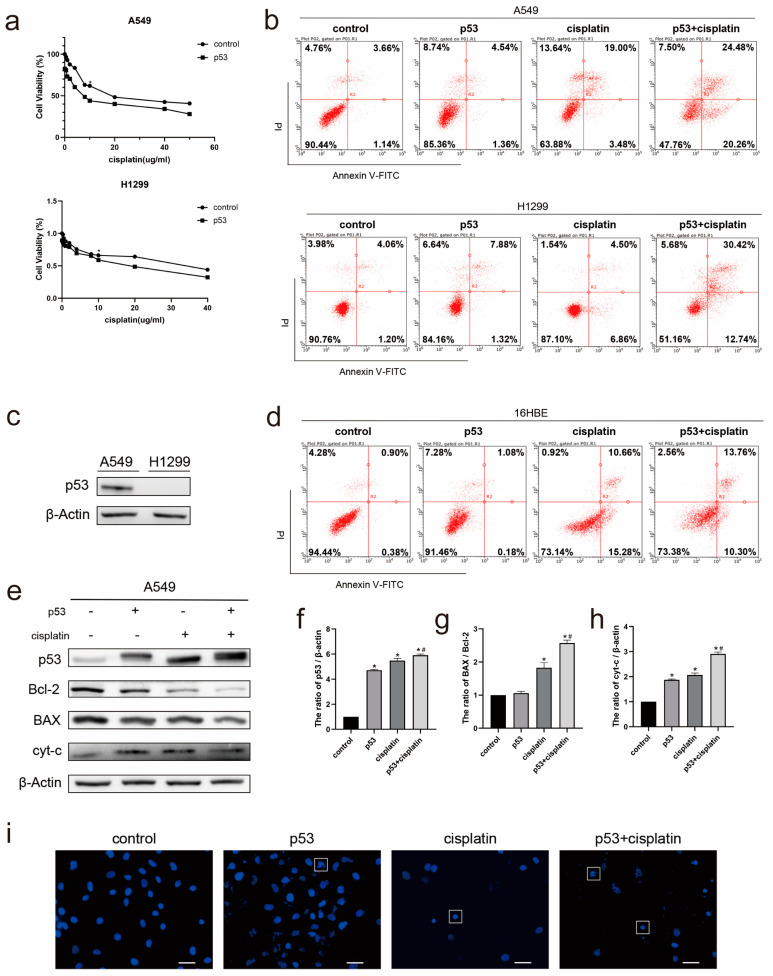
Combined treatment of p53 and cisplatin increased apoptosis in lung cancer cells. (**a**) Cell viability was determined by MTT assay. * *p* < 0.05 vs. control. (**b**) Cell apoptosis was assessed by staining with Annexin V-FITC and PI in A549 cells and H1299 cells transfected with p53 and/or treated with 10μM cisplatin for 24 h. (**c**) Western blot detection of p53 expression levels in A549 cells and H1299 cells. (**d**) Cell apoptosis was assessed by staining with Annexin V-FITC and PI in 16HBE cells transfected with p53 and/or treated with 10μM cisplatin for 24 h. (**e**) Western blot detection of apoptosis related proteins in A549 cells transfected with p53 and/or treated with 10μM cisplatin for 24 h. Quantitation of (**f**) p53, (**g**) cytochrome C levels, and (**h**) Bcl-2/Bax ratio in A549 cells. Data are presented as mean ± SD, *n* = 3. * *p* < 0.05 vs. control and # *p* < 0.05 vs. cisplatin. (**i**) A549 cells transfected with p53 and/or treated with 10μM cisplatin for 24 h and stained with Hoechst 33342. Cell morphology was observed using confocal microscopy indicate apoptotic cells. Scale bar, 20 µm.

**Figure 2 ijms-23-16034-f002:**
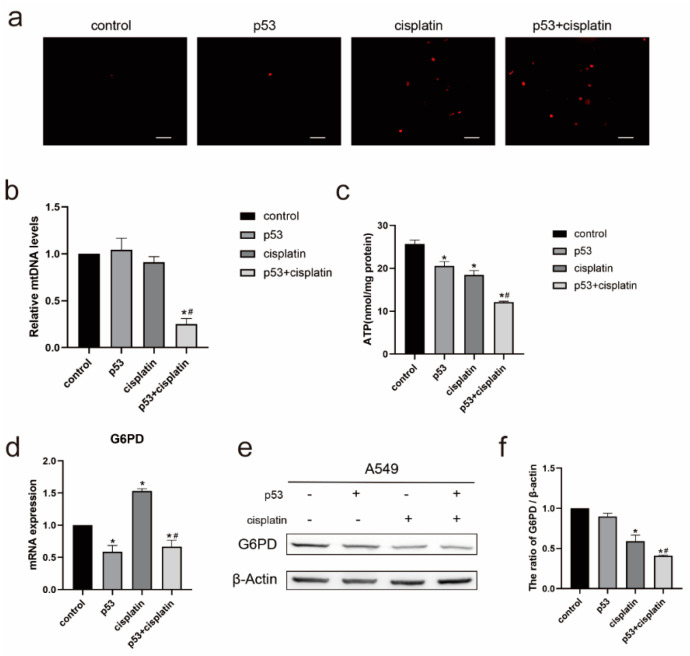
Evaluation of mitochondrial function and inhibition of the pentose phosphate pathway by overexpression of p53 and cisplatin. (**a**) The levels of mitochondrial ROS were determined using the mitochondrial superoxide indicator MitoSOX™ Red. Following transfection with p53 and/or subsequent exposure to 10μM cisplatin for 24h, ROS levels were measured. Scale bar, 20 µm. (**b**) RT-PCR detection of mitochondrial copy number in A549 cells transfected with p53 and/or subsequent exposure to 10μM cisplatin for 24h. Data are presented as mean ± SD, *n* = 3. * *p* < 0.05 vs. control and # *p* < 0.05 vs. cisplatin. (**c**) Cellular ATP levels of A549 cells transfected with p53 and/or subsequent exposure to 10μM cisplatin for 24h. Data are presented as mean ± SD, *n* = 3. * *p* < 0.05 vs. control and # *p* < 0.05 vs. cisplatin. (**d**) mRNA levels of G6PD in A549 cells transfected with p53 and/or subsequent exposure to 10μM cisplatin for 24h by RT-qPCR. Data are presented as mean ± SD, *n* = 3. * *p* < 0.05 vs. control and # *p* < 0.05 vs. cisplatin. (**e**) Western blot was used to detect G6PD protein expression in A549 cells. (**f**) The protein/actin ratio is expressed as the mean ± SD; *n* = 3, * *p* < 0.05 vs. control and # *p* < 0.05 vs. cisplatin.

**Figure 3 ijms-23-16034-f003:**
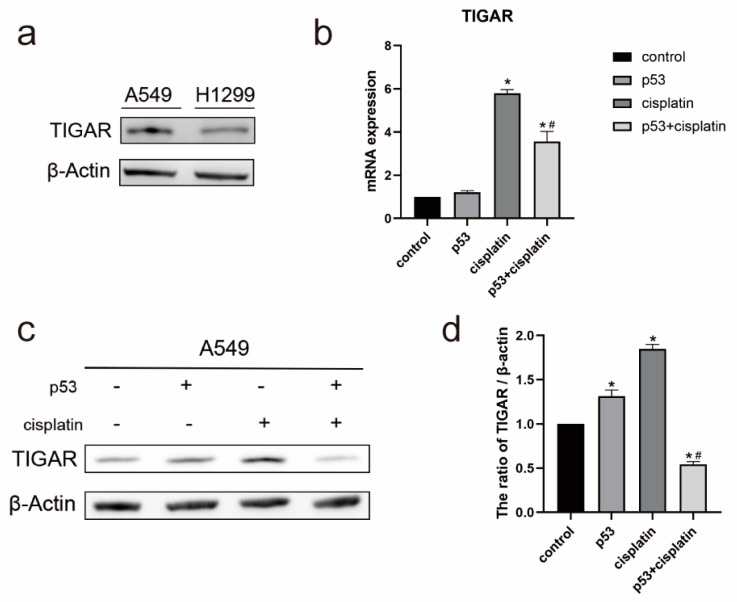
Combined treatment with p53 and cisplatin reduced TIGARs expression in A549 cells. (**a**) Western blot detection of TIGARs expression levels in A549 cells and H1299 cells. (**b**) mRNA levels of TIGAR in A549 cells transfected with p53 and/or subsequent exposure to 10μM cisplatin for 24h by RT-qPCR. Data are presented as mean ± SD, *n* = 3. * *p* < 0.05 vs. control and # *p* < 0.05 vs. cisplatin. (**c**) Western blot was used to detect TIGAR protein expression in A549 cells. (**d**) The protein/actin ratio is expressed as the mean ± SD; *n* = 3, * *p* < 0.05 vs. control and # *p* < 0.05 vs. cisplatin.

**Figure 4 ijms-23-16034-f004:**
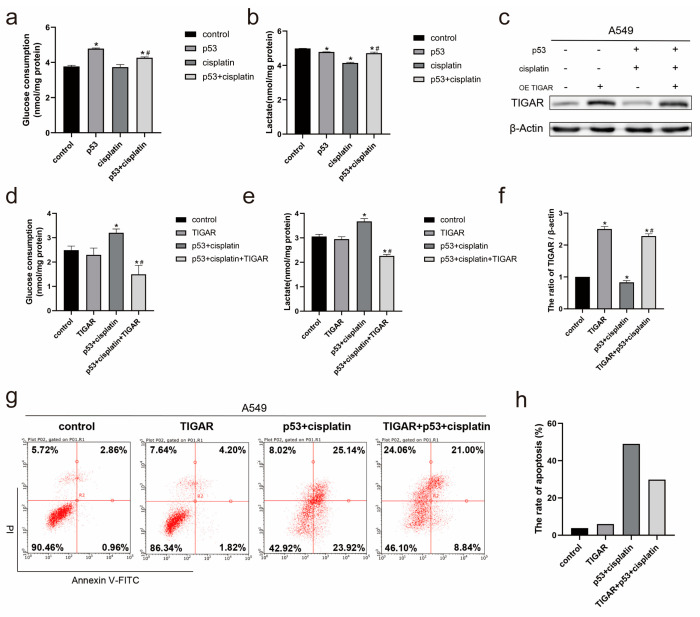
Combined treatment with p53 and cisplatin regulates the levels of glycolysis through TIGAR. (**a**) Glucose consumption and (**b**) lactate production were measured in the culture media from A549 cells transfected with p53 and/or subsequent exposure to cisplatin using glucose and lactate kit and normalized to the protein content. (**c**) Western blot was used to detect TIGAR protein expression in A549 cells. (**d**) Glucose consumption and (**e**,**f**) lactate production were measured in the culture media from A549 cells transfected with TIGAR and combined p53 and cisplatin for 24h using glucose and lactate kit and normalized to the protein content. (**g**,**h**) Cell apoptosis was assessed by staining with Annexin V-FITC and PI in A549 cells transfected with TIGAR and combined p53 and cisplatin for 24h. Data are presented as mean ± SD, *n* = 3. * *p* < 0.05 vs. control and # *p* < 0.05 vs. cisplatin.

**Figure 5 ijms-23-16034-f005:**
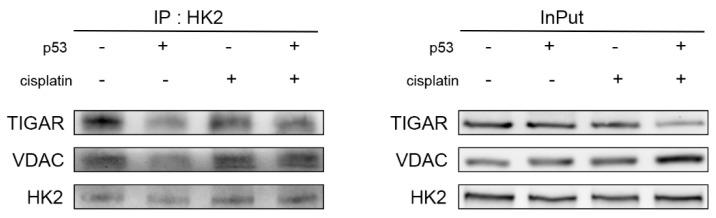
Combination treatment of cisplatin with p53 reduced the localization of the TIGAR-HK2 complex to the mitochondria. Immunoprecipitation was used to detect the binding of HK2 to TIGAR and VDAC after A549 cells were treated and expressed as the mean ± SD; *n* = 3.

**Figure 6 ijms-23-16034-f006:**
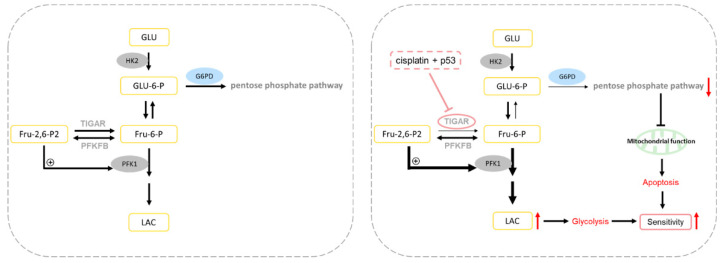
After combined treatment of p53 and cisplatin, modulation of TIGAR increased the flux of glucose to glycolysis, while the flux to the pentose phosphate pathway decreased, mitochondrial function was inhibited, and apoptosis was increased in lung cancer A549 cells, which increased sensitivity of lung cancer cells.

## Data Availability

All data are available.

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
