# Peer review of "Inhibition of TIGAR Increases Exogenous p53 and Cisplatin Combination Sensitivity in Lung Cancer Cells by Regulating Glycolytic Flux"

_ijms, 2022, doi:10.3390/ijms232416034_

Round 1

Reviewer 1 Report

This manuscript tries to provide evidence to support the conclusion that TIGAR could serve as a downstream target of p53 in the combined treatment of p53 overexpression and chemoreagent cisplatin, resulting in decreased pentose phosphate pathway during glucose metabolism, increased glucose consumption and lactate production, and increased sensitivity of lung cancer cells to cisplatin. Overall, the conclusion made by the authors could not be supported by the results they provided. 

1. The English writing (grammar problems) of this manuscript is so poor and the reasoning for this study is weird without proper logical rationalization. The manuscript should be rewritten.

2. It sounded weird when the authors deemed the experiments where cisplatin was treated in the lung cancer cells overexpressing p53 as p53 and cisplatin combined treatment. Clinically, it is impossible to transfect p53 into patients' tumor cells during or before cisplatin treatment. If they tried to enhance the function of p53, they should have used candidate reagents, e.g., small molecules or protein/antibody drugs, that end up enhancing p53 function.

3. There are numerous possibilities responsible for the increased sensitivity upon cisplatin+p53 and the regulation of glycolytic flux is only one of them. The authors did not provide evidence to corroborate that TIGAR truly mediated the altered glucose metabolism to sensitize lung cancer cells to cisplatin+p53.

4. The pathways in the illustrated paradigm in Fig. 6 are mostly speculative without enough supporting evidence.

Author Response

Response to Reviewer 1 Comments:

This manuscript tries to provide evidence to support the conclusion that TIGAR could serve as a downstream target of p53 in the combined treatment of p53 overexpression and chemoreagent cisplatin, resulting in decreased pentose phosphate pathway during glucose metabolism, increased glucose consumption and lactate production, and increased sensitivity of lung cancer cells to cisplatin. Overall, the conclusion made by the authors could not be supported by the results they provided.

We thank the reviewer for their careful and patient comments, and we will respond to each of them and revise them in the manuscript.

Point 1: The English writing (grammar problems) of this manuscript is so poor and the reasoning for this study is weird without proper logical rationalization. The manuscript should be rewritten.

Response 1: Thank you for your suggestions. We have consulted with a professional English editor and revised the manuscript.

Point 2: It sounded weird when the authors deemed the experiments where cisplatin was treated in the lung cancer cells overexpressing p53 as p53 and cisplatin combined treatment. Clinically, it is impossible to transfect p53 into patients' tumor cells during or before cisplatin treatment. If they tried to enhance the function of p53, they should have used candidate reagents, e.g., small molecules or protein/antibody drugs, that end up enhancing p53 function.

Response 2: Thanks to the reviewers for such useful suggestion. As an important tumor suppressor gene, p53 has been widely studied and used. Gene therapy of p53 has also been applied and advanced in non-small cell lung cancer. We referenced and represented in the manuscript,

Line 43-line 46, Roth et al., in 1996, used a sub-bronchial direct injection of wild-type p53 recombinant retrovirus suspension to treat nine patients with non-small-cell lung cancer who failed conventional treatment; in three cases, tumor volume decreased significantly after treatment, and in another three cases the tumor stopped growing [6].

In addition, adenoviral-mediated p53 gene transfection in the treatment of non-small cell lung cancer has also been reported,

  • Weill D, Mack M, Roth J, et al. Adenoviral-mediated p53 gene transfer to non-small cell lung cancer through endobronchial injection. 2000;118(4):966-970. doi:10.1378/chest.118.4.966
  • Swisher SG, Roth JA, Nemunaitis J, et al. Adenovirus-mediated p53 gene transfer in advanced non-small-cell lung cancer. J Natl Cancer Inst. 1999;91(9):763-771. doi:10.1093/jnci/91.9.763
  • Schuler M, Rochlitz C, Horowitz JA, et al. A phase I study of adenovirus-mediated wild-type p53 gene transfer in patients with advanced non-small cell lung cancer. Hum Gene Ther. 1998;9(14):2075-2082. doi:10.1089/hum.1998.9.14-2075
  • Liu B, Zhang H, Duan X, et al. Adenovirus-mediated wild-type p53 transfer radiosensitizes H1299 cells to subclinical-dose carbon-ion irradiation through the restoration of p53 function. Cancer Biother Radiopharm. 2009;24(1):57-66. doi:10.1089/cbr.2008.0514

In this study, we hope to know whether p53 gene and common chemotherapeutic drug cisplatin can play a synergistic anti-tumor effect in lung cancer cells, and what the mechanism is. We therefore used p53 and cisplatin combined treatment to investigate the molecular mechanisms underlying this synergy to prove our hypothesis.

Meanwhile, we agree with your idea that in clinical treatment, transfection of p53 cannot be widely used by current medical methods, the biggest obstacle for clinical application is the lack of an effective, non-immunogenic delivery system. In recent years, studies have reported that new drug delivery systems are expected to achieve safer and more efficient p53 delivery methods, for example:

  • Zou Y, Tornos C, Qiu X, Lia M, Perez-Soler R. p53 aerosol formulation with low toxicity and high efficiency for early lung cancer treatment. Clin Cancer Res. 2007;13(16):4900-4908. doi:10.1158/1078-0432.CCR-07-0395
  • Munagala R, Aqil F, Jeyabalan J, et al. Exosome-mediated delivery of RNA and DNA for gene therapy. Cancer Lett. 2021;505:58-72. doi:10.1016/j.canlet.2021.02.011
  • Choi SH, Jin SE, Lee MK, et al. Novel cationic solid lipid nanoparticles enhanced p53 gene transfer to lung cancer cells. Eur J Pharm Biopharm. 2008;68(3):545-554. doi:10.1016/j.ejpb.2007.07.011

these methods still need further research and verification. Therefore, in-depth research on the molecular mechanism of gene therapy is particularly important, which can provide a theoretical basis for clinical treatment. In addition, the application of small molecule or protein /antibody drugs will be presented in our later experiments. Thanks again for your suggestion.

Point 3: There are numerous possibilities responsible for the increased sensitivity upon cisplatin+p53 and the regulation of glycolytic flux is only one of them. The authors did not provide evidence to corroborate that TIGAR truly mediated the altered glucose metabolism to sensitize lung cancer cells to cisplatin+p53.

Response 3: We agree with the reviewer that regulation of glycolytic flux is only one possibility for increasing drug sensitivity. Previous work in our laboratory has demonstrated that modulating glucose metabolism increases drug sensitivity, and We have supplemented this part to the manuscript.

Xu Y, Gao W, Zhang Y, et al. ABT737 reverses cisplatin resistance by targeting glucose metabolism of human ovarian cancer cells. Int J Oncol. 2018;53(3):1055-1068. doi:10.3892/ijo.2018.4476

Xiang XY, Kang JS, Yang XC, et al. SIRT3 participates in glucose metabolism interruption and apoptosis induced by BH3 mimetic S1 in ovarian cancer cells. Int J Oncol. 2016;49(2):773-784. doi:10.3892/ijo.2016.3552

The regulatory role of TIGAR in glycolytic flux is summarized as follows,

Line 61-line 71, The protein structure of TIGAR is highly similar to the structural sequence of 2,6-bisphosphofructokinase in 6-phosphofructokinase-2/2,6-bisphosphofructokinase (PFK-2/FBPase-2), which can hydrolyze intracellular fructose 2,6-bisphosphate. Fructose 2,6-bisphosphate substrate is a strong activator of phosphofructokinase-1 (PFK-1), since TIGAR is able to hydrolyze intracellular fructose 2,6-bisphosphate to fruc-tose-6-phosphate; the activity of PFK-1 is reduced by this, which in turn inhibits the cellular glycolytic pathway. At the same time, TIGAR can also regulate the pentose phosphate pathway (PPP), so that fructose 6-phosphate enters the pentose phosphate pathway to promote the synthesis of nicotinamide adenine dinucleotide phosphate (NADPH) and 5-phosphoribonucleic acid, thereby exerting an antioxidant effect and reducing reactive oxygen species (ROS)-induced apoptosis.

Green DR, Chipuk JE. p53 and metabolism: Inside the TIGAR. Cell. 2006;126(1):30-32. doi:10.1016/j.cell.2006.06.032

Therefore, in this study, we examined glucose consumption, lactate production, and G6PD which the first rate-limiting enzyme of the pentose phosphate pathway to demonstrate altered glucose metabolism.

Point 4: The pathways in the illustrated paradigm in Fig. 6 are mostly speculative without enough supporting evidence.

Response 4: Thank you for your suggestions. We have supplemented the manuscript with the relevant mechanisms involved in pathways in the illustrated paradigm.

Line 259-line 265, Additionally, previous work in our lab has also demonstrated that modulation of glycolysis can increase sensitivity to chemotherapeutics, for example, Xu et al. demonstrated that ABT737 reversed cisplatin resistance by targeting glucose metabolism in human ovarian cancer cells, and Xiang et al. demonstrated that S1 exerts anticancer effects in SKOV3 cells by interrupting glucose metabolism and inducing apoptosis involving activation of SIRT3, while the combination of glycolysis inhibitor 2-DG and S1 is able to further enhance cytotoxicity.

Line 282-line 287, Veronica et al. believed that mitochondria were involved in cisplatin resistance and were related to mitochondrial DNA, mitochondrial dynamics, and mitophagy. For example, Wandee et al. showed chemosensitizing activity of metformin in combination with cisplatin in cholangiocarcinoma, an effect associated with increased oxidative stress associated with mitochondrial dysfunction and initiation of cell death.

We would like to thank the reviewer again for taking the time to review our manuscript.

Reviewer 2 Report

Glycolysis and mitochondrial respiration play significantly in tumorigenesis and metastatic formation. Study of their mechanism and regulation is highly requested. Nevertheless, the submitted manuscript has irreparable flags within the review process. Unfortunately, I have no choice but to recommend its rejection.

Authors primary clam, activation TIGAR is possible strategy for induction of sensitivity of A549 to combination of p53 and cisplatin therapy. Nevertheless, Figure 3b and c (page 5) showed, that in this case of combination, is expression significantly lower against control and single agents.  

Also, previous works (you can see below) strongly imply, TIGAR activation is very probably part adaptive mechanism against chemical and agents and radioactivity.

Hsieh SL, Chen CT, Wang JJ, Kuo YH, Li CC, Hsieh LC, Wu CC. Sedanolide induces autophagy through the PI3K, p53 and NF-κB signaling pathways in human liver cancer cells. Int J Oncol. 2015 Dec;47(6):2240-6. doi: 10.3892/ijo.2015.3206. Epub 2015 Oct 15. PMID: 26500073.

Sinha S, Ghildiyal R, Mehta VS, Sen E. ATM-NFκB axis-driven TIGAR regulates sensitivity of glioma cells to radiomimetics in the presence of TNFα. Cell Death Dis. 2013 May 2;4(5):e615. doi: 10.1038/cddis.2013.128. PMID: 23640457; PMCID: PMC3674344.

Zhao M, Fan J, Liu Y, Yu Y, Xu J, Wen Q, Zhang J, Fu S, Wang B, Xiang L, Feng J, Wu J, Yang L. Oncogenic role of the TP53-induced glycolysis and apoptosis regulator in nasopharyngeal carcinoma through NF-κB pathway modulation. Int J Oncol. 2016 Feb;48(2):756-64. doi: 10.3892/ijo.2015.3297. Epub 2015 Dec 17. PMID: 26691054.

Based above works, it can be expected correlation between activity TIGAR and NF-kB. It is well known, that cytostatic such as diplatinum can activated NF-kB and its signalling can repress mitochondrial apoptotic pathway. Observed change in the mitochondrial ROS level may not be related at all TIGAR expression and localization.

Your interpretation of cited results is completely wrong. From example, line 287- In addition, studies have shown that TIGAR increases the sensitivity of various chemotherapy drugs such as cisplatin, 5-FU and temozolomide [21,23,31].

Ref. 21 Tang Z, He Z. TIGAR promotes growth, survival and metastasis through oxidation resistance and AKT activation in glioblastoma. Oncol Lett. 2019 Sep;18(3):2509-2517. doi: 10.3892/ol.2019.10574. Epub 2019 Jul 5. PMID: 31402948; PMCID: PMC6676722.

From the abstract: Glioblastoma has a poor prognosis and is one of the most lethal types of cancer in the world. TP53 induced glycolysis regulatory phosphatase (TIGAR) is upregulated in various types of cancer. The results of the present study revealed that TIGAR was positively associated with poor survival and was upregulated in glioblastoma. TIGAR knockdown significantly increased oxidative stress, decreased cell proliferation and exacerbated TMZ-induced apoptosis in U-87MG cells. Additionally, TIGAR knockdown decreased migration, invasion and EMT, and treatment of TIGAR-shRNA-transfected cells with NADPH had no effect on metastasis. In addition, TIGAR promoted AKT activation and bound to AKT. In conclusion, the present study demonstrated that TIGAR may promote glioblastoma growth and progression through oxidation resistance and AKT activation.

Ref. 23 Huang, S.; Yang, Z.; Ma, Y.; Yang, Y.; Wang, S. miR-101 Enhances Cisplatin-Induced DNA Damage Through Decreasing Nicotinamide Adenine Dinucleotide Phosphate Levels by Directly Repressing Tp53-Induced Glycolysis and Apoptosis Regulator Expression in Prostate Cancer Cells. DNA and Cell Biology 2017, 36, 303-310, doi:10.1089/dna.2016.3612.

From the abstract: Furthermore, this study revealed that the roles of knockdown of TIGAR were similar to miR-101 upregulation in prostate cancer cells. Taken together, miR-101 inhibited viability, induced apoptosis, reprogramed glucose metabolism, and enhanced cisplatin-induced DNA damage through decreasing NADPH levels by directly suppressing the expression of TIGAR in prostate cancer cells.

Ref. 31 Chu, J.; Niu, X.; Chang, J.; Shao, M.; Peng, L.; Xi, Y.; Lin, A.; Wang, C.; Cui, Q.; Luo, Y.; et al. Metabolic remodeling by TIGAR overexpression is a therapeutic target in esophageal squamous-cell carcinoma. Theranostics 2020, 10, 3488-3502, 478 doi:10.7150/thno.41427.

From the abstract: We show that TP53-induced glycolysis and apoptosis regulator (TIGAR) is a major player in ESCC progression and chemoresistance. TIGAR reprograms glucose metabolism from glycolysis to the glutamine pathway through AMP-activated kinase, and its overexpression is correlated with poor disease outcomes. Tigar knockout mice have reduced ESCC tumor burden and growth rates. Treatment of TIGAR-overexpressing ESCC cell xenografts and patient-derived tumor xenografts in mice with combination of glutaminase inhibitor and chemotherapeutic agents achieves significant more efficacy than chemotherapy alone.

Author Response

Response to Reviewer 2 Comments:

Glycolysis and mitochondrial respiration play significantly in tumorigenesis and metastatic formation. Study of their mechanism and regulation is highly requested. Nevertheless, the submitted manuscript has irreparable flags within the review process. Unfortunately, I have no choice but to recommend its rejection.

We thank the reviewer for their careful and patient comments, and sorry for making you mistake the conclusion of this article, we will respond to each of comments and revise them in the manuscript.

Point 1: Authors primary clam, activation TIGAR is possible strategy for induction of sensitivity of A549 to combination of p53 and cisplatin therapy. Nevertheless, Figure 3b and c (page 5) showed, that in this case of combination, is expression significantly lower against control and single agents. 

Response 1: Thank you for your evaluation. We must apologize for the misunderstanding we caused you due to the inappropriate use of words in the title, and revise the title to:

Inhibition of TIGAR increases the sensitivity of p53 and cisplatin combination in lung cancer A549 cells by regulating glycolytic flux.

We primary clam, reduction of TIGAR expression after combined treatment of p53 and cisplatin could serve as a target to increase the sensitivity of A549 cells. This is consistent with what is described in the manuscript.

Line 255-line 257, In this study, we found that the reduction in TIGAR after combined treatment of p53 and cisplatin altered the metabolic pattern of tumor cells,

Line 323-line 325, In conclusion, we found that p53 combined with cisplatin treatment reduced the expression of TIGAR, the pentose phosphate pathway was inhibited, the glucose flux shifted more towards glycolysis, and the metabolic pattern of A549 cells changed.

In addition, Figure 3b and c (page 5) showed, that in this case of combination, is expression significantly lower against control and single agents corresponds to our conclusions. We have carefully checked and confirmed the manuscript according to the reviewer's comments.

Point 2: Also, previous works (you can see below) strongly imply, TIGAR activation is very probably part adaptive mechanism against chemical and agents and radioactivity.

Hsieh SL, Chen CT, Wang JJ, Kuo YH, Li CC, Hsieh LC, Wu CC. Sedanolide induces autophagy through the PI3K, p53 and NF-κB signaling pathways in human liver cancer cells. Int J Oncol. 2015 Dec;47(6):2240-6. doi: 10.3892/ijo.2015.3206. Epub 2015 Oct 15. PMID: 26500073.

Sinha S, Ghildiyal R, Mehta VS, Sen E. ATM-NFκB axis-driven TIGAR regulates sensitivity of glioma cells to radiomimetics in the presence of TNFα. Cell Death Dis. 2013 May 2;4(5):e615. doi: 10.1038/cddis.2013.128. PMID: 23640457; PMCID: PMC3674344.

Zhao M, Fan J, Liu Y, Yu Y, Xu J, Wen Q, Zhang J, Fu S, Wang B, Xiang L, Feng J, Wu J, Yang L. Oncogenic role of the TP53-induced glycolysis and apoptosis regulator in nasopharyngeal carcinoma through NF-κB pathway modulation. Int J Oncol. 2016 Feb;48(2):756-64. doi: 10.3892/ijo.2015.3297. Epub 2015 Dec 17. PMID: 26691054.

Based above works, it can be expected correlation between activity TIGAR and NF-kB. It is well known, that cytostatic such as diplatinum can activated NF-kB and its signalling can repress mitochondrial apoptotic pathway. Observed change in the mitochondrial ROS level may not be related at all TIGAR expression and localization.

Response 2: The comments made by the reviewers are valuable and we are glad to answer them. Related studies can be referenced.

  • Jiang LB, Cao L, Ma YQ, et al. TIGAR mediates the inhibitory role of hypoxia on ROS production and apoptosis in rat nucleus pulposus cells. Osteoarthritis Cartilage. 2018;26(1):138-148. doi:10.1016/j.joca.2017.10.007

In this study the authors demonstrated that TIGAR has a protective effect on ROS production, and after silencing TIGAR, both mitochondrial and total cellular ROS levels were increased in hypoxia-treated NP cells. This indicated that TIGAR silence reversed the inhibitory effects of hypoxia on intracellular and mitochondrial ROS production.

  • Cheung EC, Ludwig RL, Vousden KH. Mitochondrial localization of TIGAR under hypoxia stimulates HK2 and lowers ROS and cell death. Proc Natl Acad Sci U S A. 2012;109(50):20491-20496. doi:10.1073/pnas.1206530109

In this study the authors have shown that TIGAR can decrease mitochondria ROS during hypoxia when HK2 is present on the mitochondria.

  • Liu Z, Wu Y, Zhang Y, et al. TIGAR Promotes Tumorigenesis and Protects Tumor Cells From Oxidative and Metabolic Stresses in Gastric Cancer. Front Oncol. 2019;9:1258. Published 2019 Nov 19. doi:10.3389/fonc.2019.01258

This study’s data showed that TIGAR knockdown significantly increased the ROS production and reduced the generation of NADPH, leading to reduced NADPH/NADP+ ratios, elevated mitochondrial ATP production, and phosphorus oxygen ratios.

Point 3: Your interpretation of cited results is completely wrong. From example, line 287- In addition, studies have shown that TIGAR increases the sensitivity of various chemotherapy drugs such as cisplatin, 5-FU and temozolomide [21,23,31].

Ref. 21 Tang Z, He Z. TIGAR promotes growth, survival and metastasis through oxidation resistance and AKT activation in glioblastoma. Oncol Lett. 2019 Sep;18(3):2509-2517. doi: 10.3892/ol.2019.10574. Epub 2019 Jul 5. PMID: 31402948; PMCID: PMC6676722.

From the abstract: Glioblastoma has a poor prognosis and is one of the most lethal types of cancer in the world. TP53 induced glycolysis regulatory phosphatase (TIGAR) is upregulated in various types of cancer. The results of the present study revealed that TIGAR was positively associated with poor survival and was upregulated in glioblastoma. TIGAR knockdown significantly increased oxidative stress, decreased cell proliferation and exacerbated TMZ-induced apoptosis in U-87MG cells. Additionally, TIGAR knockdown decreased migration, invasion and EMT, and treatment of TIGAR-shRNA-transfected cells with NADPH had no effect on metastasis. In addition, TIGAR promoted AKT activation and bound to AKT. In conclusion, the present study demonstrated that TIGAR may promote glioblastoma growth and progression through oxidation resistance and AKT activation.

Ref. 23 Huang, S.; Yang, Z.; Ma, Y.; Yang, Y.; Wang, S. miR-101 Enhances Cisplatin-Induced DNA Damage Through Decreasing Nicotinamide Adenine Dinucleotide Phosphate Levels by Directly Repressing Tp53-Induced Glycolysis and Apoptosis Regulator Expression in Prostate Cancer Cells. DNA and Cell Biology 2017, 36, 303-310, doi:10.1089/dna.2016.3612.

From the abstract: Furthermore, this study revealed that the roles of knockdown of TIGAR were similar to miR-101 upregulation in prostate cancer cells. Taken together, miR-101 inhibited viability, induced apoptosis, reprogramed glucose metabolism, and enhanced cisplatin-induced DNA damage through decreasing NADPH levels by directly suppressing the expression of TIGAR in prostate cancer cells.

Ref. 31 Chu, J.; Niu, X.; Chang, J.; Shao, M.; Peng, L.; Xi, Y.; Lin, A.; Wang, C.; Cui, Q.; Luo, Y.; et al. Metabolic remodeling by TIGAR overexpression is a therapeutic target in esophageal squamous-cell carcinoma. Theranostics 2020, 10, 3488-3502, 478 doi:10.7150/thno.41427.

From the abstract: We show that TP53-induced glycolysis and apoptosis regulator (TIGAR) is a major player in ESCC progression and chemoresistance. TIGAR reprograms glucose metabolism from glycolysis to the glutamine pathway through AMP-activated kinase, and its overexpression is correlated with poor disease outcomes. Tigar knockout mice have reduced ESCC tumor burden and growth rates. Treatment of TIGAR-overexpressing ESCC cell xenografts and patient-derived tumor xenografts in mice with combination of glutaminase inhibitor and chemotherapeutic agents achieves significant more efficacy than chemotherapy alone.

Response 3: We apologize for such errors, and have corrected them in the revised manuscript.

Line 314-line 316, In addition, studies have shown that knockdown of TIGAR increases the sensitivity of various chemotherapy drugs such as cisplatin, 5-FU and temozolomide [21,25,38].

We would like to thank the reviewer again for taking the time to review our manuscript.

Reviewer 3 Report

In this paper, authors investigated the regulation of TIGAR by the combination therapy of p53 and cisplatin plays an important role in increasing the sensitivity of tumor cells to cisplatin. They found that combined treatment of p53 and cisplatin was able to inhibit mitochondrial function, promote mitochondrial pathway-induced apoptosis and increase sensitivity in A549 cells. However, this paper is not enough to clearly explaining these mechanisms.

Major

1.      As you know, the p53 tumor-suppressor protein prevents cancer development through various mechanisms, including the induction of cell-cycle arrest, apoptosis, and the maintenance of genome stability. In this study, overexpression of p53 may be increased the cisplatin-induced apoptosis in lung cancer cells?. In the Figure 1, cisplatin treated A549 cells increased the expression of p53 and thus what is the difference of combination therapy after added p53?

2.      Authors should be confirmed the basal expression of p53 and TIGAR expression in 549 cells.

3.      Abbreviation clearly corrected : Hexokinase 2 (HK2)

4.      As you indicated in Figure 1, p53 expression was increased and apoptotic cell population was also increased, how about the expression levels of p21?

5.      Figure 5. The resolution of western blot band is very poor. Authors changed this figures.

6.      Authors should be measured and compared the OCR and ECAR.

7.      Authors measured the levels of fructose-2,6-biphosphate because By lowering F-2,6-P2 levels, TIGAR decreases the activity of PFK-1 and reduces glycolytic flux downstream of this point. 

8.      As you know, mutant p53s generally lose the ability to activate wild-type p53 target genes, they retain the ability to control transcription, such as the activation of genes involved in the mevalonate pathway. As TIGAR expression is preserved in tumor cells that carry mutations in p53, it is possible that some mutant p53s retain the ability to influence the expression of TIGAR and so help to promote tumorigenesis. Therefore, authors might be compared the p53 types in this study.

Author Response

Response to Reviewer 3 Comments:

In this paper, authors investigated the regulation of TIGAR by the combination therapy of p53 and cisplatin plays an important role in increasing the sensitivity of tumor cells to cisplatin. They found that combined treatment of p53 and cisplatin was able to inhibit mitochondrial function, promote mitochondrial pathway-induced apoptosis and increase sensitivity in A549 cells. However, this paper is not enough to clearly explaining these mechanisms.

We thank the reviewer for their careful and patient comments, and we will respond to each of them and revise them in the manuscript.

Major

Point 1: As you know, the p53 tumor-suppressor protein prevents cancer development through various mechanisms, including the induction of cell-cycle arrest, apoptosis, and the maintenance of genome stability. In this study, overexpression of p53 may be increased the cisplatin-induced apoptosis in lung cancer cells?. In the Figure 1, cisplatin treated A549 cells increased the expression of p53 and thus what is the difference of combination therapy after added p53?

Response 1: Yes, overexpression of p53 increased the cisplatin-induced apoptosis in A549 cells. In the experiments of Karim et al., which regarding the identification of TIGAR as a p53-inducible gene, endogenous p53 activation was assessed by treatment of wild-type p53-expressing U2OS or RKO cells with actinomycin D.

Bensaad K, Tsuruta A, Selak MA, et al. TIGAR, a p53-inducible regulator of glycolysis and apoptosis. Cell. 2006;126(1):107-120. doi:10.1016/j.cell.2006.05.036

And the transfection of p53 is to overexpress p53 in cells through plasmids. We speculate that this combination therapy might be considered beneficial in null p53 or mutant p53 tumor cells that lose tumor suppressor function, since such tumor cells cannot activate endogenous p53.

Point 2: Authors should be confirmed the basal expression of p53 and TIGAR expression in 549 cells.

Response 2: Thank you for your suggestion. In this study we aimed to investigate the molecular mechanisms of p53 and TIGAR in metabolism and apoptosis in A549 cells, rather than their clinical expression in cancer. We would like to take your suggestion into account in our next research, thank you again for your suggestion.

Point 3: Abbreviation clearly corrected : Hexokinase 2 (HK2)

Response 3: Thank you very much for your reminder. We have checked and modified them in the manuscript.

Point 4: As you indicated in Figure 1, p53 expression was increased and apoptotic cell population was also increased, how about the expression levels of p21?

Response 4: It is a pleasure to reply to your comment. p21, a well-established cyclin-dependent kinase inhibitor, was found to play an important role in controlling cell cycle progression. p21 has been mostly associated with p53 protein regarding its cell cycle arrest role. However, our research hopes to use TIGAR as a target to explore the mechanism of energy metabolism and apoptosis and drug sensitivity. Thanks to the reviewers for such useful suggestion. We are willing to consider whether p21 and TIGAR can act as dual targets to increase the sensitivity of chemotherapeutic drugs in our follow-up experiments.

Point 5: Figure 5. The resolution of western blot band is very poor. Authors changed this figures.

Response 5: We are sorry for the poor resolution of the western blot bands due to our error, and we used the original image with unadjusted brightness in Figure5.

Point 6: Authors should be measured and compared the OCR and ECAR.

Response 6: Thank you very much for pointing out this important issue. We agree with your opinion. It is important to measure and compare OCR and ECAR to further demonstrate changes in glucose flow. However, we need time to prepare the cells and may not get the final data in the given time. Thanks again for your suggestion, we may test this part in our follow-up experiments.

Point 7: Authors measured the levels of fructose-2,6-biphosphate because By lowering F-2,6-P2 levels, TIGAR decreases the activity of PFK-1 and reduces glycolytic flux downstream of this point.

Response 7: Thank you for your evaluation. We have explained in the introduction about the function of TIGAR and fructose-2,6-biphosphate in glycolysis,

Line 61-line 67, The protein structure of TIGAR is highly similar to the structural sequence of 2,6-bisphosphofructokinase in 6-phosphofructokinase-2/2,6-bisphosphofructokinase (PFK-2/FBPase-2), which can hydrolyze intracellular fructose 2,6-bisphosphate. Fructose 2,6-bisphosphate substrate is a strong activator of phosphofructokinase-1 (PFK-1), since TIGAR is able to hydrolyze intracellular fructose 2,6-bisphosphate to fruc-tose-6-phosphate; the activity of PFK-1 is reduced by this, which in turn inhibits the cellular glycolytic pathway.

Point 8: As you know, mutant p53s generally lose the ability to activate wild-type p53 target genes, they retain the ability to control transcription, such as the activation of genes involved in the mevalonate pathway. As TIGAR expression is preserved in tumor cells that carry mutations in p53, it is possible that some mutant p53s retain the ability to influence the expression of TIGAR and so help to promote tumorigenesis. Therefore, authors might be compared the p53 types in this study.

Response 8: The comments made by the reviewers are valuable and we are glad to answer them. We agree with the reviewer on whether different types of p53 regulate TIGAR differently, and we think this is an interesting point that should be explored in depth. We have supplemented this part to the manuscript.

Line 303-line 314, As a transcription factor, the type of p53 should also be explored. p53 is frequently mutated in human tumors. In addition to the loss of original tumor-suppressive function, tumor-associated mutant p53 proteins often acquire new tumorigenic activities. In metabolic regulation, wild-type p53 inhibits glycolysis, whereas mutant p53 promotes glycolysis through different mechanisms [35]. Zhang et al. [36] found that the inhibition of glycolysis in tumor cells greatly impaired the role of mutant p53 in promoting tumorigenesis. Additionally, Jiang et al. [37] reported that tumor-associated p53 mutants lack G6PD inhibitory activity and that enhanced PPP glucose flux due to p53 inactivation may increase glucose consumption and direct glucose to biosynthesis in tumor cells. Therefore, whether mutant p53 has the ability to affect TIGAR expression, thereby promoting tumorigenesis, should be considered and investigated. This contributes to developing our understanding of the mechanism of action of TIGAR in tumor metabolism.

We would like to thank the reviewer again for taking the time to review our manuscript.

Round 2

Reviewer 1 Report

The authors did not rewrite the manuscript and design more appropriate experiments to make the reasoning more understandable and acceptable. 

1. Reportedly, TIGAR, a target of p53, is upregulated by p53 (also shown in Fig. 3). Cisplatin increases p53 expression (Fig. 1c), consequently enhancing TIGAR expression (Fig. 3). Logically, there is no reason to combine p53 overexpression and cisplatin to initiate the study. It should have been more reasonable if the authors used a tumor cell line that is totally resistant to cisplatin and expresses an extremely low level of p53. In such a case, it is reasonably expected to see a synthetic lethality when combining p53 and cisplatin (each treatment alone does not cause cell death).

2. While combining cisplatin with p53 did not result in much increased p53 expression (as compared to each treatment alone) and apoptosis in A549 cells, the combined treatment synergistically reduced mitochondrial function and pentose phosphate pathway. Such inconsistency uncouples the causal relationship between both.

3. Moreover, without performing additional experiments to explain the seemingly discrepant results in Fig. 3 where treatment of p53 or cisplatin alone increased TIGAR, whereas combined treatment drastically decreased it, the underlying mechanism for the effects of combined treatment in this manuscript is far from clear.

4. Throughout the manuscript, the authors only presented some inconsistent phenomena for combining p53 and cisplatin but did not try to design appropriate experiments at all to mechanistically connect and resolve the causal relationship in supporting the conclusion made in the title of this manuscript.

Author Response

Response to Reviewer 1 Comments

The authors did not rewrite the manuscript and design more appropriate experiments to make the reasoning more understandable and acceptable.

We thank the reviewer for their careful and patient comments. We will respond to each of them and we have revised the manuscript.

  1. Reportedly, TIGAR, a target of p53, is upregulated by p53 (also shown in Fig. 3). Cisplatin increases p53 expression (Fig. 1c), consequently enhancing TIGAR expression (Fig. 3). Logically, there is no reason to combine p53 overexpression and cisplatin to initiate the study. It should have been more reasonable if the authors used a tumor cell line that is totally resistant to cisplatin and expresses an extremely low level of p53. In such a case, it is reasonably expected to see a synthetic lethality when combining p53 and cisplatin (each treatment alone does not cause cell death).

Thanks to the reviewers for such useful suggestion. In this study, we hope that the combined treatment of p53 and cisplatin can achieve a similar effect to cisplatin, thereby reducing the dosage of cisplatin. Although cisplatin is used as a first-line chemotherapy drug, its long-term side effects are also very large. Then, the treatment method combined with p53 can achieve the same therapeutic effect by reducing the dosage, thereby reducing the side effects of cisplatin to patients. At the same time, we also very much agree with your point of view, and in the future, we will also find and use tumor cell lines that is totally resistant to cisplatin and expresses an extremely low level of p53, continue to research and validate the opinion of.

  1. While combining cisplatin with p53 did not result in much increased p53 expression (as compared to each treatment alone) and apoptosis in A549 cells, the combined treatment synergistically reduced mitochondrial function and pentose phosphate pathway. Such inconsistency uncouples the causal relationship between both.

Thank you for your suggestions. Regarding the results of apoptosis, during the experiment, the cells were washed three times after staining and centrifuged, which may lead to the loss of dead cells, and the results of apoptosis did not appear to increase significantly. However, the results of western showed that the increase of BAX/Bcl-2 ratio and the significant increase of cytochrome C release indicated that the combined treatment of p53 and cisplatin would lead to apoptosis.

  1. Moreover, without performing additional experiments to explain the seemingly discrepant results in Fig. 3 where treatment of p53 or cisplatin alone increased TIGAR, whereas combined treatment drastically decreased it, the underlying mechanism for the effects of combined treatment in this manuscript is far from clear.

We must admit that we are not yet able to explain the underlying mechanism of TIGAR reduction after combination therapy, but we did come to this result. We hope that our report can attract more people's attention to jointly conduct research on this phenomenon, and we will continue to carry out further research and verification in the future.

  1. Throughout the manuscript, the authors only presented some inconsistent phenomena for combining p53 and cisplatin but did not try to design appropriate experiments at all to mechanistically connect and resolve the causal relationship in supporting the conclusion made in the title of this manuscript.

Thank you for your evaluation. In this study, we demonstrate that combined treatment of p53 and cisplatin reduces TIGAR expression. As TIGAR is able to inhibit glycolysis and regulate glucose flux towards the pentose phosphate pathway, therefore, we examined glucose consumption, lactate production, and G6PD which the first rate-limiting enzyme of the pentose phosphate pathway to demonstrate altered glucose metabolism. In addition, we also verified the reduction of mitochondrial function, which provides the basis for increasing cisplatin sensitivity. But we very much agree with you that there are some underlying mechanisms that need to be explored further, which we will demonstrate in our future experiments.

We would like to thank the reviewer again for taking the time to review our manuscript.

Reviewer 2 Report

Current version of the manuscript it starts to make sense. Nevertheless, authors should discuss possible limitations of presented model, for example difference between protein and mRNA level (Fig. 3) and possible role of NF-kB in the mitochondrial apoptotic pathway.

Author Response

Response to Reviewer 2 Comments

Current version of the manuscript it starts to make sense. Nevertheless, authors should discuss possible limitations of presented model, for example difference between protein and mRNA level (Fig. 3) and possible role of NF-kB in the mitochondrial apoptotic pathway.

It is an honor to receive such comments from reviewers. Gene expression is affected by various regulatory mechanisms such as transcriptional level regulation, post-transcriptional regulation, and post-translational regulation. Therefore, the protein and mRNA levels may be consistent or different. In this study, the functional role of TIGAR and its protein expression are more important, which we will add in the discussion. (Line 287-line 289)

In addition, our main objective in this study was to investigate TIGAR regulation of glycolytic flux and decrease mitochondrial function to provide evidence for increased cisplatin sensitivity. But we agree with the reviewer about NF-kB and think it is an interesting point that should be explored in depth, which we will demonstrate in our future experiments.

We would like to thank the reviewer again for taking the time to review our manuscript.

Reviewer 3 Report

The authors did not clearly corrected and added experiments according to the reviewer's comments.  

Author Response

Response to Reviewer 3 Comments

The authors did not clearly corrected and added experiments according to the reviewer's comments.

We thank the reviewer for their comments.

  1. As you know, the p53 tumor-suppressor protein prevents cancer development through various mechanisms, including the induction of cell-cycle arrest, apoptosis, and the maintenance of genome stability. In this study, overexpression of p53 may be increased the cisplatin-induced apoptosis in lung cancer cells?. In the Figure 1, cisplatin treated A549 cells increased the expression of p53 and thus what is the difference of combination therapy after added p53?

Yes, overexpression of p53 increased the cisplatin-induced apoptosis in A549 cells. Combination therapy after added p53 can achieve a similar therapeutic effect to cisplatin treatment with a reduced dose of cisplatin. Although cisplatin is used as a first-line chemotherapy drug, its long-term side effects are also very large. Then, the treatment method combined with p53 can achieve the same therapeutic effect by reducing the dosage, thereby reducing the side effects of cisplatin to patients.

  1. Authors should be confirmed the basal expression of p53 and TIGAR expression in 549 cells.

Thank you for your suggestion. We have confirmed the basal expression of p53 and TIGAR expression shows in Figure 1 and Figure 3.

  1. Abbreviation clearly corrected : Hexokinase 2 (HK2)

Thank you very much for your reminder. We have checked and modified them in the manuscript.

  1. As you indicated in Figure 1, p53 expression was increased and apoptotic cell population was also increased, how about the expression levels of p21?

Thank you for your suggestion. Our group's previous studies on apoptosis and cell cycle demonstrated that the expression level of p21 increased after overexpression of p53. Since this study focused on the glucose metabolism regulated by TIGAR, we did not detect p21 here.

  1. Figure 5. The resolution of western blot band is very poor. Authors changed this figures.

We are sorry for the poor resolution of the western blot bands due to our error, and we used the original image with unadjusted brightness in Figure5.

  1. Authors should be measured and compared the OCR and ECAR.

Thank you for your suggestion. We regret that due to the limited experimental equipment conditions in our laboratory, we are currently unable to supplement these experiments at the given time.

  1. Authors measured the levels of fructose-2,6-biphosphate because By lowering F-2,6-P2 levels, TIGAR decreases the activity of PFK-1 and reduces glycolytic flux downstream of this point.

Thank you for your suggestion. We hope to supplement this experiment, but we are very sorry that due to the epidemic, we are currently unable to purchase a fructose 2,6-bisphosphate detection kit for the experiment.

  1. As you know, mutant p53s generally lose the ability to activate wild-type p53 target genes, they retain the ability to control transcription, such as the activation of genes involved in the mevalonate pathway. As TIGAR expression is preserved in tumor cells that carry mutations in p53, it is possible that some mutant p53s retain the ability to influence the expression of TIGAR and so help to promote tumorigenesis. Therefore, authors might be compared the p53 types in this study.

The comments made by the reviewers are valuable and we are glad to answer them. We agree with the reviewer on whether different types of p53 regulate TIGAR differently, and we think this is an interesting point that should be explored in depth. We have supplemented this part to the manuscript.

As a transcription factor, the type of p53 should also be explored. p53 is frequently mutated in human tumors. In addition to the loss of original tumor-suppressive function, tumor-associated mutant p53 proteins often acquire new tumorigenic activities. In metabolic regulation, wild-type p53 inhibits glycolysis, whereas mutant p53 promotes glycolysis through different mechanisms [35]. Zhang et al. [36] found that the inhibition of glycolysis in tumor cells greatly impaired the role of mutant p53 in promoting tumorigenesis. Additionally, Jiang et al. [37] reported that tumor-associated p53 mutants lack G6PD inhibitory activity and that enhanced PPP glucose flux due to p53 inactivation may increase glucose consumption and direct glucose to biosynthesis in tumor cells. Therefore, whether mutant p53 has the ability to affect TIGAR expression, thereby promoting tumorigenesis, should be considered and investigated. This contributes to developing our understanding of the mechanism of action of TIGAR in tumor metabolism.

In addition, the p53 background of the cell line A549 used in this experiment is wild-type p53, which has also been supplemented in the manuscript.

We would like to thank the reviewer again for taking the time to review our manuscript.

Round 3

Reviewer 1 Report

The response from the authors to my second comment #1 stated that combining treatment with p53 and a lower dosage of cisplatin can reduce the side effects of cisplatin on patients. However, there was no evidence whatsoever supporting such a statement. How do they know that p53+cisplatin does not cause the same (or even worse) side effects?

The response from the authors to my second comment #2 is simply speculative without showing evidence supporting what they argued. They should have redone the apoptosis assay without washing out the staining dyes (the flow cytometer only reads cells but does not read the unstained dyes, which could easily be differentiated by gating with higher FSC/SSC cells) to prove their speculation. The results from apoptosis assays presented the end point of cell death but not BAX/Bcl-2 ratio and cytochrome C release.

Overall, even though the authors agreed with all of my second comments, they have no intention AT ALL of performing additional experiments to support the conclusion made in this manuscript's title and improve its biological and scientific soundness. Their response is not satisfactory.

Author Response

Response to Reviewer 1 Comments

The response from the authors to my second comment #1 stated that combining treatment with p53 and a lower dosage of cisplatin can reduce the side effects of cisplatin on patients. However, there was no evidence whatsoever supporting such a statement. How do they know that p53+cisplatin does not cause the same (or even worse) side effects?

We reperformed apoptosis assay and the combined treatment group showed a higher apoptosis ratio. This could demonstrate that the combination of low dose cisplatin can achieve the effect of high dose cisplatin alone.

The response from the authors to my second comment #2 is simply speculative without showing evidence supporting what they argued. They should have redone the apoptosis assay without washing out the staining dyes (the flow cytometer only reads cells but does not read the unstained dyes, which could easily be differentiated by gating with higher FSC/SSC cells) to prove their speculation. The results from apoptosis assays presented the end point of cell death but not BAX/Bcl-2 ratio and cytochrome C release.

Thank you for your suggestion. We have redone the apoptosis assay shows in Figure 1b. In addition, the BAX/Bcl-2 ratio and cytochrome C release shows in Figure 1d.

Overall, even though the authors agreed with all of my second comments, they have no intention AT ALL of performing additional experiments to support the conclusion made in this manuscript's title and improve its biological and scientific soundness. Their response is not satisfactory.

We thank the reviewers for their careful and patient comments and are sorry for not making you understand our points.

In this study, we first examined the cell apoptosis. And the results of relevant proteins of apoptosis in mitochondrial pathway suggested that mitochondrial pathway apoptosis is predominant. Subsequently, we demonstrated that an impairment of mitochondrial function occurred. The pentose phosphate pathway was also examined. Since TIGAR serves as a target gene for p53 and is able to modulate the pentose phosphate pathway, we focused on the changes in TIGAR at the mRNA and protein levels. We next examined the content of glucose and lactate to validate the function of TIGAR. Finally, we examined the interaction of TIGAR and HK2 on the mitochondria, further demonstrating the role of TIGAR in increasing sensitivity.

We must apologize for the misunderstanding we caused you and could you please explicitly the additional experiments that should be performed. Meanwhile, we have revised the title to: Exogenous p53 inhibit TIGAR and regulate glycolytic flux increases cisplatin sensitivity in lung cancer cells.

We would like to thank the reviewer again for taking the time to review our manuscript.

Reviewer 3 Report

This paper clearly revised again according to the reviewer's comments. 

Author Response

Response to Reviewer 3 Comments

This paper clearly revised again according to the reviewer's comments.

We would like to thank the reviewer again for taking the time to review our manuscript and make careful and patient suggestions.

Round 4

Reviewer 1 Report

The authors misunderstood the first part of my previous comment. Here, I restate my concern again: “The response from the authors to my second comment #1 stated that combining treatment with p53 and a lower dosage of cisplatin can reduce the side effects of cisplatin on patients. However, there was no evidence whatsoever supporting such a statement. How do they know that p53+cisplatin does not cause the same (or even worse) side effects?”. After redoing the apoptosis assay in new Fig. 1, I have no doubt about the higher apoptosis ratio in the combined treatment group. But I was concerned about the combined treatment's side effects on cancer patients. The authors showed no evidence of a possible reduction of side effects.

I did not misunderstand the authors’ points. I just cannot accept that decreased TIGAR expression is responsible for the increased sensitivity of the tumor cells to cisplatin combined with the treatment of exogenous p53 if they do not do additional experiments to prove their hypothesis. Specifically, they should test whether restoring TIGAR expression (e.g., ectopically expressing TIGAR) in the tumor cells treated with exogenous p53 and cisplatin rescue the cell viability, decrease the mitochondrial pathway apoptosis, and decrease the level of glycolysis. The authors should also go back to my second comment for the details about the additional experiments that should be performed.

The revised title is Grammarly incorrect and less clear than the previous one.

Author Response

Response to Reviewer 1 Comments

The authors misunderstood the first part of my previous comment. Here, I restate my concern again: “The response from the authors to my second comment #1 stated that combining treatment with p53 and a lower dosage of cisplatin can reduce the side effects of cisplatin on patients. However, there was no evidence whatsoever supporting such a statement. How do they know that p53+cisplatin does not cause the same (or even worse) side effects?”. After redoing the apoptosis assay in new Fig. 1, I have no doubt about the higher apoptosis ratio in the combined treatment group. But I was concerned about the combined treatment's side effects on cancer patients. The authors showed no evidence of a possible reduction of side effects.

    We thank the reviewer for their careful and patient comments. Please allow me to explain the results in Figure1. The rate of apoptosis increased after the combination treatment compared to cisplatin alone, therefore, we believe that the use of this combination therapy method, even if reduced the amount of cisplatin, can achieve the effect of cisplatin treatment alone. In addition, the feasibility of using exogenous p53 have mentioned in first response #2, therefore, we believe that reducing the amount of cisplatin through combination therapy may reduce the side effects.

The first response #2: [ Line 43-line 46, Roth et al., in 1996, used a sub-bronchial direct injection of wild-type p53 recombinant retrovirus suspension to treat nine patients with non-small-cell lung cancer who failed conventional treatment; in three cases, tumor volume decreased significantly after treatment, and in another three cases the tumor stopped growing [6].

In addition, adenoviral-mediated p53 gene transfection in the treatment of non-small cell lung cancer has also been reported,

â‘     Weill D, Mack M, Roth J, et al. Adenoviral-mediated p53 gene transfer to non-small cell lung cancer through endobronchial injection. Chest. 2000;118(4):966-970. doi:10.1378/chest.118.4.966

â‘¡    Swisher SG, Roth JA, Nemunaitis J, et al. Adenovirus-mediated p53 gene transfer in advanced non-small-cell lung cancer. J Natl Cancer Inst. 1999;91(9):763-771. doi:10.1093/jnci/91.9.763

â‘¢    Schuler M, Rochlitz C, Horowitz JA, et al. A phase I study of adenovirus-mediated wild-type p53 gene transfer in patients with advanced non-small cell lung cancer. Hum Gene Ther. 1998;9(14):2075-2082. doi:10.1089/hum.1998.9.14-2075

â‘£    Liu B, Zhang H, Duan X, et al. Adenovirus-mediated wild-type p53 transfer radiosensitizes H1299 cells to subclinical-dose carbon-ion irradiation through the restoration of p53 function. Cancer Biother Radiopharm. 2009;24(1):57-66. doi:10.1089/cbr.2008.0514 ]

I did not misunderstand the authors’ points. I just cannot accept that decreased TIGAR expression is responsible for the increased sensitivity of the tumor cells to cisplatin combined with the treatment of exogenous p53 if they do not do additional experiments to prove their hypothesis. Specifically, they should test whether restoring TIGAR expression (e.g., ectopically expressing TIGAR) in the tumor cells treated with exogenous p53 and cisplatin rescue the cell viability, decrease the mitochondrial pathway apoptosis, and decrease the level of glycolysis. The authors should also go back to my second comment for the details about the additional experiments that should be performed.

    We thank the reviewers for their careful and patient comments. According to reviewer’s comment we added experiments to test whether restoring TIGAR expression in the tumor cells treated with exogenous p53 and cisplatin rescue the cell viability and decrease the level of glycolysis, and the results have shown in Figure4.

    Line 205-line 212, In addition, we increased TIGAR expression by transfection plasmid (Fig. 4c) and treated cisplatin and p53 transfection for 24 hours in A549 cells with increased TIGAR expression levels, the glucose consumption (Fig. 4d) and lactate production (Fig. 4e) in were both reduced after the increase of TIGAR expression. Moreover, flow cytometry experiments showed that the increased TIGAR expression reversed the level of apoptosis due to the combination therapy in A549 cells treated with combined p53 and cis-platin (Fig. 4g-h). We believe that the co-treatment of p53 with cisplatin changes the metabolic pattern of cells by the downregulation of the TIGAR expression and increased the cell apoptosis.

The revised title is Grammarly incorrect and less clear than the previous one.

    Thank you for your suggestion. We have chosen the previous title: Inhibition of TIGAR increases exogenous p53 and cisplatin combination sensitivity in lung cancer cells by regulating glycolytic flux.

    We would like to thank the reviewer again for taking the time to review our manuscript.

Round 5

Reviewer 1 Report

In terms of the side effects that the combination of p53 and cisplatin might cause, what I exactly concerned about was whether combining drugs causes the same or even worse adverse side effects, e.g., cell apoptosis, than the drug alone on normal cell types. The authors should additionally test, at least, the apoptotic effects of control, p53 alone, cisplatin alone, and p53+cisplatin on normal cells, such as normal lung endothelial or epithelial cell line, and make sure the drug combination does not enhance the apoptosis of the normal cell lines as compared to that of cells treated with cisplatin or p53 alone.

It is good that the authors were finally willing to perform some of the requested experiments to demonstrate the essential role of TIGAR in the combining effects of cisplatin and exogenous p53 expression. However, according to the new Fig.4c, 4g, and 4h, overexpression TIGAR did not restore the apoptotic effects back to the control level. Does it mean other molecules than TIGAR might be responsible for the combining apoptotic effect? The authors should address such inconsistency in the Discussion section.

Author Response

Response to Reviewer 1 Comments

In terms of the side effects that the combination of p53 and cisplatin might cause, what I exactly concerned about was whether combining drugs causes the same or even worse adverse side effects, e.g., cell apoptosis, than the drug alone on normal cell types. The authors should additionally test, at least, the apoptotic effects of control, p53 alone, cisplatin alone, and p53+cisplatin on normal cells, such as normal lung endothelial or epithelial cell line, and make sure the drug combination does not enhance the apoptosis of the normal cell lines as compared to that of cells treated with cisplatin or p53 alone.

    We thank the reviewers for their careful and patient comments. According to reviewer’s comment we added flow cytometry experiments to test the apoptotic effects of control, p53 alone, cisplatin alone, and p53+cisplatin on normal cells (16HBE cells), and the results have shown in Figure 1d.

Line 109 - line113: In addition, we performed the same treatment in 16HBE cell lines of bronchial epithelial cells, and the flow cytometry results showed that compared with cisplatin treatment, the combination treatment of p53 and cisplatin did not cause further apoptosis of 16HBE cells (Fig.1d).

It is good that the authors were finally willing to perform some of the requested experiments to demonstrate the essential role of TIGAR in the combining effects of cisplatin and exogenous p53 expression. However, according to the new Fig.4c, 4g, and 4h, overexpression TIGAR did not restore the apoptotic effects back to the control level. Does it mean other molecules than TIGAR might be responsible for the combining apoptotic effect? The authors should address such inconsistency in the Discussion section.

    Thank you for your suggestion. The increased expression of TIGAR can reverse the apoptosis by the combination of p53 and cisplatin, which demonstrated the role of TIGAR on apoptosis after the combination treatment. But the increased expression of TIGAR did not reverse all apoptosis, we also believe that there may be other mechanisms regulating apoptosis after combined treatment, which is worthy of further study in the future. And we add this to the discussion.

Line 304 - line 307: Our study confirmed that the increased expression of TIGAR can reverse the apoptosis by the combination of p53 and cisplatin, but this reversal is limited, and there may be other mechanisms that need to be explored for the regulation of the combination therapy on lung cancer cell apoptosis.

    We would like to thank the reviewer again for taking the time to review our manuscript.

Round 6

Reviewer 1 Report

With the authors' proper revision, I have only one final concern remaining. In Fig. 1d, the doses of p53 and cisplatin used for normal cells should be the same as those for tumor cells for a fair comparison. Moreover, according to the apoptosis level of the control cells, the basal apoptotic effect was a little too high, unexpectedly resulting in no effect of cisplatin on normal cells. On the contrary, cisplatin+p53 treatment was even more apoptotic than cisplatin alone, likely making a wrong suggestion that the side effect of cisplatin+p53 treatment was more severe than cisplatin alone. Therefore, the experiment should be redone by adding cisplatin up to 10uM (instead of 5uM) cisplatin for 24 h. But be sure that the apoptosis of control should not be too high.

Author Response

Response to Reviewer 1 Comments

With the authors' proper revision, I have only one final concern remaining. In Fig. 1d, the doses of p53 and cisplatin used for normal cells should be the same as those for tumor cells for a fair comparison. Moreover, according to the apoptosis level of the control cells, the basal apoptotic effect was a little too high, unexpectedly resulting in no effect of cisplatin on normal cells. On the contrary, cisplatin+p53 treatment was even more apoptotic than cisplatin alone, likely making a wrong suggestion that the side effect of cisplatin+p53 treatment was more severe than cisplatin alone. Therefore, the experiment should be redone by adding cisplatin up to 10uM (instead of 5uM) cisplatin for 24 h. But be sure that the apoptosis of control should not be too high.

    We thank the reviewers for their careful and patient comments. According to reviewer’s comment we reperformed flow cytometry experiments, and changed the dose of cisplatin(10uM), and the results have shown in Figure 1d. The results showed that the combination treatment of p53 and cisplatin did not cause further apoptosis compared with cisplatin alone treatment.

    We would like to thank the reviewer again for taking the time to review our manuscript.

Round 7

Reviewer 1 Report

The authors have properly responded to my comments and revised the manuscript. I have no further questions.